# Inhibition of Wdr5 Attenuates Ang-II-Induced Fibroblast-to-Myofibroblast Transition in Cardiac Fibrosis by Regulating Mdm2/P53/P21 Pathway

**DOI:** 10.3390/biom12111574

**Published:** 2022-10-27

**Authors:** Jiali Yuan, Hong Peng, Binfeng Mo, Chengye Yin, Guojian Fang, Yingze Li, Yuepeng Wang, Renhua Chen, Qunshan Wang

**Affiliations:** 1Department of Cardiology, Xinhua Hospital, School of Medicine, Shanghai Jiao Tong University, #1665 Kongjiang Road, Shanghai 200082, China; 2Department of Cardiology, Quanzhou Hospital of Traditional Chinese Medicine, #388 SunJiang Road, Quanzhou 362000, China

**Keywords:** cardiac fibrosis, Wdr5, H3K4me3, epigenetics, cell senescence

## Abstract

Cardiac fibrosis is an important pathological process in many diseases. Wdr5 catalyzes the trimethylation of lysine K4 on histone H3. The effects of Wdr5 on the cardiac fibrosis phenotype and the activation or transformation of cardiac fibroblasts were investigated by Ang-II-infused mice by osmotic mini-pump and isolated primary neonatal rat cardiac fibroblasts. We found that the Wdr5 expression and histone H3K4me3 modification were significantly increased in Ang-II-infused mice. By stimulating primary neonatal rat cardiac fibroblasts with Ang II, we detected that the expression of Wdr5 and H3K4me3 modification were also significantly increased. Two Wdr5-specific inhibitors, and the lentivirus that transfected Sh-Wdr5, were used to treat primary mouse cardiac fibroblasts, which not only inhibited the histone methylation by Wdr5 but also significantly reduced the activation and migration ability of Ang-II-treated fibroblasts. To explore its mechanism, we found that the inhibition of Wdr5 increased the expression of P53, P21. Cut&Tag-qPCR showed that the inhibition of Wdr5 significantly reduced the enrichment of H3K4me3 in the Mdm2 promoter region. For in vivo experiments, we finally proved that the Wdr5 inhibitor OICR9429 significantly reduced Ang-II-induced cardiac fibrosis and increased the expression of P21 in cardiac fibroblasts. Inhibition of Wdr5 may mediate cardiac fibroblast cycle arrest through the Mdm2/P53/P21 pathway and alleviate cardiac fibrosis.

## 1. Introduction

Cardiovascular diseases are the number one disease-related deaths worldwide. Cardiac fibrosis plays a pivotal role in a wide variety of chronic and cardiovascular diseases such as heart failure, atrial fibrillation, remodeling of hypertension, and hypertrophy [1,2]. Fibrosis of the heart is an adaptive response of the heart to various external stimuli and is the hallmark of the hearts of aged people. Cardiac fibroblasts are central actors in depositing collagens and other extracellular matrices and are responsible for cardiac fibrosis [3]. During cardiac fibrosis, activated Angiotensin II (Ang II), transforming growth factor-β (TGF-β) signaling, and mechanosensing mediate the accumulation of fibroblasts and fibroblast-to-myofibroblast transition (FMT) [3,4]. Myofibroblasts are specialized fibroblasts that express the contractile protein α-smooth muscle actin (α-SMA) and appears in the heart after injury and as the heart ages. Numerous previous studies reported an increased number of activated fibroblasts in aged myocardium or injured hearts [5,6,7].

In the aged heart, expansion of interstitial and perivascular fibrosis, ECM deposition, and fibroblasts starting to change in phenotypes are major differences [4,8]. Nevertheless, it is found that fibroblasts derived from aging hearts had reduced responsiveness to TGF-β, showing defects in TGF-β pathways [4,9]. Previous research proved that the induction of senescence of selective fibroblasts is associated with decreased myocardial interstitial fibrosis and helped improve cardiac function [10,11,12,13]. Therefore, fibroblast senescence may be associated with healthy cardiac aging.

Epigenetic dysregulation plays a crucial role in the process of cardiac fibrosis; for example, by regulating gene expression via the acetylation and methylation of histone. We previously found that the increased activity of the histone methyltransferase enhancer of zeste homolog 2 (EZH2) and EZH2-mediated-H3K27me3 are partially responsible for atrial fibroblast differentiation and atrial fibrosis [14,15]. Liu et al. proved the relationship between two critical histone marks, H3K27me3 and H3K4me3, in cardiac fibroblast [16]. We wonder whether H3K4me3 is also crucial in regulating fibroblast-to-myofibroblast transition.

It is generally admitted that trithorax group (TrxG) complexes (the SWI/SNF complex and COMPASS family) are responsible for the trimethylation of lysine 4 on histone H3(H3K4me3) [17]. A well-conserved protein core of all COMPASS complexes among all creatures is WD domain repeat 5 (Wdr5). Wdr5 is an extraordinarily highly conserved protein among vertebrates [18]. Numerous studies showed that Wdr5 promotes cellular differentiation and embryo formation [19,20]. It plays a central scaffolding role in COMPASS complexes via its two key binding domains, which is critical for robust histone methyltransferases (HMT) activity [21]. Most of the studies proved that suppressed Wdr5 may be a new therapeutic target in cancers [22,23]. Inhibition of Wdr5 and its methyltransferase activity attenuated renal fibrosis and inflammation in the kidney with ischemia-reperfusion injury [24], and it is involved in renal fibrosis in diabetic nephropathy [25]. Displacement and inhibition of Wdr5 generally caused impeded translation, nucleolar stress, and induction of p53-dependent apoptosis and ceased cell cycle [26,27]. Altogether, these findings led us to the hypothesis that Wdr5 may be a key regulator in cardiac fibrosis, suppressed Wdr5 and its HMT activity is a new approach to treat cardiac fibrosis.

Accordingly in our study, we show that Wdr5 and H3K4me3 is upregulated in Ang-ΙΙ-treated mice for 4 weeks. Pharmacologic and genetic inhibitions of Wdr5 suppresses Ang-II-induced FMT in vitro and Ang-II-induced cardiac fibrosis in vivo. Wdr5-mediated H3K4me3 modification is enriched at the promoter of Mdm2. Finally, we demonstrate the inhibition of Wdr5 is associated with the Mdm2/p53/p21 pathway, and finally, induced cell senescence. Overall, our data suggest that Ang-II-induced H3K4me3 is a candidate target for cardiac fibrosis.

## 2. Materials and Methods

### 2.1. Animal Studies

Animal studies were approved by the Animal Care and Use Committee of Shanghai Xinhua Hospital affiliated to Shanghai Jiao Tong University School to Medicine. Eight-week-old C57/B6 mice (male, bodyweight: 23 ± 2 g) were anesthetized intraperitoneally using 3% sodium pentobarbital (50 mg/kg). The adequacy of anesthesia was confirmed by the absence of reflex response to foot squeeze. An osmotic mini-pump (Alzet 2004, Cupertino, CA, USA) was implanted for subcutaneous infusion of Angiotensin II (Ang II, Sigma, Shanghai, China, A9525, 1000 ng/kg/min) for 4 weeks to induce cardiac fibrosis according to our previous study [14]. Ten mice received Ang II infusion and ten mice in the control group received Saline infusion. To validate the effect of cardiac fibrosis, we used another agonist-induced cardiac fibrosis model. We exposed ten mice to continuous injection of isoproterenol hydrochloride (ISO, MCE, Shanghai, China, #HY-B0468, 50 mg/kg dissolved in sterile saline) for 14 days as described previously [28]. We observed no difference in mortality (the mortality was 0% for all groups) or infection after the treatment compared to WT littermates. Before the mice were sacrificed, echocardiography and blood pressure investigations were performed. The hearts were extracted on the 28th day after Ang II infusion or the 14th day after ISO (50 mg/kg) injection subcutaneously. At the end of the experiments, mice were killed with the intraperitoneal injection of an overdose of sodium pentobarbital (200 mg/kg), and the death was confirmed by cervical dislocation.

For in vivo inhibition of Wdr5, six mice received an i.p. injection of OICR-9429 at a dose of 3 mg/kg per day before the mice were sacrificed.

### 2.2. Echocardiographic Measurements

On the 28th day, post-Ang II or saline infusion echocardiography was performed with an ultrasound instrument (Vevo 3100, FUJIFILM, Shanghai, China) and the transducer was MS400 (30 MHz). Mice were anesthetized with 2% isoflurane inhalation. To maintain a 37 °C body temperature, the mice were placed on a heated pad. LA size was assessed by the parasternal long-axis view and was measured at end-ventricular systole.

### 2.3. Isolation and Culture of Neonatal Rat Cardiac Fibroblasts

Neonatal rat cardio fibroblasts were isolated from 1-day-old neonatal Wistar rats. After sacrifice, hearts were excised and flushed with D-Hanks to get rid of any remaining blood. The ventricles were cut into 1 mm^3^ pieces and washed by D-Hanks twice. Then, the fragments were transferred to a tube containing 0.125% trypsin and shaken for 15 min at 37 °C. After that, the fragments were washed twice by D-Hanks and re-suspended at about 10 mL DMEM (low glucose) and supplanted with 1% collagenase type I (Worthington, LS004194). The tissue and the suspensions were shaken in 37 °C water until no obvious tissue pieces were left (approximately 1.5–2 h) to separate cardiomyocytes and fibroblasts. After that, the cell suspensions were combined, filtered using sterile, and plated on uncoated cultured dishes for 1 h. After 1 h, the adhered cells were collected and plated on culture dishes. After 24 h, the medium was replaced to remove the dead cell. Fibroblasts were maintained in high glucose DMEM medium supplemented with 10% FBS and 1% penicillin/streptomycin (P/S). The medium was changed every other day.

For different stimulation, after changing the medium, cells were treated with or without 1 µM or 5 µM Ang II (MCE, HY-13948) for 48 h. 

### 2.4. Western Blotting and Quantitative Real-Time qPCR

Proteins from the cardiac ventricular tissue of mice and cultured fibroblasts were extracted by RIPA that contained protease inhibitors and phosphatase inhibitors. An equal quantity of protein (20–40 μg) was resolved by SDS/PAGE and transferred to PVDF membranes. The blots were blocked with 5% non-fat milk in TBS buffer with 0.5% Tween-20 for at least 2 h at room temperature and then incubated with antibodies against Wdr5 (1:1000, Abcam, ab178410), H3K4me3 (1:1500, Abcam, ab8580), α-SMA (1:3000, Abcam, ab6276), Collagen I (1:800, Abclonal, A16891), P53 (1:1000, Abclonal, A5761), Mdm2 (1:1000, Abclonal, A0345), P21 (1:800, Abcam, ab109199), and α-tubulin (1:2000, Abcam, ab7291). Then, the blots were incubated with primary antibody at 4 °C overnight (about 12 h). The blots were then incubated with secondary antibodies conjugated with horseradish peroxidase (1:1000, Beyotime, A0208,) for 1–2 h at room temperature. The Gel Imaging System (Tanon) and Image J software were used to image and analyze the intensity of each band normalized to loading control α-tubulin.

Total RNA was extracted from cardiac ventricular tissues and cultured cells using TRIzol (Takara, Cat# RR036A). Total RNA (1000 ng) was reversely transcribed into cDNA using the Prime-ScriptTMRT reagent kit (Takara, RR036A). qRT-PCR was performed using SYBR green (Yeasen, 11199ES03) and normalized to GAPDH expression. ABI QuantStudio 3 (Applied Biosystems) with standard PCR conditions (95 °C for 30 s, followed by 40 cycles of 95 °C for 10 s and 60 °C for 30 s) was used to run the samples. The sequences of the primers (synthesized by Sangon Biotech, Shanghai, China) for the target genes are shown in Appendix A.

### 2.5. Wdr5 Inhibition by Small Molecules and Knockdown

NRCFs were cultured in DMEM medium supplemented with 10% FBS and 1% P/S. The purified fibroblasts were then passed on to 6-well culture plates at a density of 6 × 10^5^ cells/well or culture slide. Preincubation with OICR-9429 (TargetMol, T6916) (5 μmol/L) or MM-401 (S7265; Selleck) (50 μmol/L) was performed for 24 h before Ang II stimulation in NRCFs.

For knockdown of Wdr5 in cardiac fibroblasts of neonatal rats, a specific hairpin sequence sh-Wdr5 was subcloned into the lentivirus. Cells were infected with the lentivirus (500 MOI) for 48 h, followed by replacement of fresh serum-free media for another 24 h. Cells were then used for WB validation, migration assay, and Ang II stimulation study. The targeting sequence of the small interfering RNA: (1) GCAAGTTCCTCTGCTGATA (2) GCTCATTGATGACGACAAT (3) CCAGTCTCAGCCGTTCATT.

### 2.6. EdU Staining

The 5-ethynyl-2’-deoxyuridine (EdU) assay was conducted to evaluate the proliferation of NRCFs following the manufacturer’s protocols of the BeyoClick™ (Shanghai, China) EdU Cell Proliferation Kit (Beyotime, C0071S) with Alexa Fluor 647. In brief, cultured NRCFs were incubated in 10 mM EdU solution for 2 h. After fixation, EdU in cells was labeled by fluorescent probes, which can be detected under an inverted fluorescence microscope. Quantification of EdU staining compared to total nucleus number (DAPI stained) was performed using ImageJ.

### 2.7. Propidium Iodide Staining and Flow Cytometry

Cells were trypsinized and suspended with 70% ethanol at 4 °C for 2 h. After washing twice with ice-cold PBS, cells were incubated with propidium iodide staining solution (50 μg/mL) in dark for 10 min. After being transferred to FACS tubes, cells were delivered to an EPICS XL flow cytometer (Beckman Coulter, CA, USA). The Cell Quest software was used to analyze the results.

### 2.8. Transwell Migration Assay and Scratching Assay

An 8 μm pore size insert was initially placed in a 24-well plate (Corning, 3422), and primary rat cardiac fibroblasts were then seeded into wells at a density of 20,000/mL in serum-free medium. Twenty-four hours later, all cells inside the insert were removed, only the migrated cells on the outside were washed with PBS, fixed with 4% formalin, stained with 0.25% crystal violet for 15 min, rinsed with sterile water, allowed to dry, and finally mounted. Five visual fields of 200× magnification were collected to count the number of migrating cells in each field.

For a scratching assay, fibroblasts were plated at a density of 5.0 × 10^4^/mL well in 6-well plates. A single scratch was made by a sterile 200 µL pipette tip when cells reached 90% confluence. We drew different marks on the bottom of the plates to ensure that the cells in the same visual field were serially assayed to migrate. Quantification of the migration speed was measured using ImageJ 1.8.0 software.

### 2.9. β-Gal-Staining

SA-β-gal staining was performed by a staining kit according to the manufacturer’s protocol (Cell Signaling Technology, 9860S). Briefly, different groups NRCFs were washed with PBS and fixed using the fixative solution for half an hour at room temperature and then incubated at 37 °C overnight with the SA-β-gal staining solution. The senescent NRCFs stained with blue were photographed. The percentage was calculated from five different view fields of each sample in three independent experiments.

### 2.10. Cellular Immunofluorescence

Cultured cells were prepared as described previously. After different stimulations, primary rat cardiac fibroblasts were fixed in 4% paraformaldehyde for 15 min and permeabilized with 0.1% TritonX100 in PBS for 10 min. After blocking with 5% BSA for 1 h at room temperature, the cells were incubated with primary antibody α-SMA (1:200, Abcam, ab5694) or H3K4me3(1:400, CST, #9751) at 4 °C overnight. The cells were then incubated with DAPI and Alexa Fluor ^®^ 488-conjugated goat anti-rabbit IgG H&L (Beyotime, A0423) at room temperature for 1 h. The cells were visualized and photographed under an Olympus IX71 microscope (Olympus, Tokyo, Japan).

### 2.11. Histology

The cardiac tissue was excised from anesthetized wild-type (WT) and Ang II-treated mice. The process of paraformaldehyde-fixing and paraffin-embedding was previously described [14]. Sections (5 μm thick) of atrial or ventricular tissue were deparaffinized and stained with Masson’s trichrome staining kit (HT15-1KT; Sigma-Aldrich, Shanghai, China) or for IHC or tissue immunofluorescence. The IHC or Masson’s staining was visualized and photographed under an Olympus BX51 microscope (Olympus, Tokyo, Japan). ImageJ was used to analyze quantitative cardiac fibrosis only in ventricular fibrosis areas and the atrial areas are excluded.

### 2.12. Tissue Immunofluorescence

Heart tissues from mice were fixed with 10% phosphate-buffered formalin for 24 h. Fixed tissues were then paraffin-embedded and serially sectioned with a microtome (4 μm thickness). Immunofluorescence vimentin and P21 staining of heart sections were performed using mouse anti-vimentin antibody (Abcam, ab8978, 1:250) and rabbit anti-P21 antibody (Abcam, ab188224, 1:100). Then, the sectioned tissues were incubated with secondary antibodies (Alexa Fluor^®^ 488-conjugated goat anti-rabbit IgG H&L (Beyotime, Shanghai, China, A0423) and Alexa Fluor^®^ 647-conjugated goat anti-mouse IgG H&L (Beyotime, Shanghai, China, A0473) and DAPI were used to stain the cell nuclei (blue). The expression of vimentin and P21 in each mouse was the mean value of three slices in three different sections. 

### 2.13. Lentivirus Assembly and Transduction 

psPAX2, pMD2.G, and modified MV-MCS-EF1α-copGFP-T2A-puro carrying coding sequence of Mdm2 were co-transfected with the ratio of 4:2:1, into Hek-293T, packaging host cell line, by using polyethylenimine (PEI; Polysciences) with the ratio of 3:1, PEI (μg): total DNA (μg). The supernatant medium was refreshed 24 h after transfection. The supernatant recombinant lentivirus-green particles were collected at 48 and 72 h, subsequent to the transfection. The lentivirus was collected and concentrated by Universal Virus Concentration Kit (Beyotime, C2901). Afterward, the NRCFs were transduced by the recombinant lentivirus particles accompanying the 8μg/mL polybrene. After 2 days of incubation, the NRCFs were harvested.

### 2.14. Cut&Tag Assay

The Cut&Tag is a new method to extract the DNA bound to protein. A group of NRCF cells was treated with lentiviruses of sh-NC or sh-Wdr5 and another group of NRCF cells was treated with DMSO or OICR-9429 (5 μmol/L) in 6-well plates. Then, the cells were harvested and treated according to the manufacturer’s instructions by using in situ DNA Binding Library for Illumina Cut&Tag kits (Yeasen, 12598E12). The DNAs were extracted and subjected to PCR and qPCR and the DNA spike-in was used as a reference. The information on the rat Mdm2 promoter primers and DNA spike-in are listed in Appendix A. A detailed step-by-step protocol can be found at: https://www.yeasen.com/products/detail/1819 (accessed on 10 October 2021).

### 2.15. Data Analysis

Data were analyzed using Graph-Pad prism 9.0 or SPSS 19.0 statistical software and represented as mean ± SEM or percentage of at least four independent experiments. Two-tailed Student’s tests were used for two-group comparisons, and ANOVA followed by post hoc Tukey’s test was used for multiple-group comparisons. A value of *p* < 0.05 was considered statistically significant.

## 3. Results

### 3.1. Upregulation of Fibrosis and Wdr5 in Different Animal Models

To explore the crucial gene in developing cardiac fibrosis, we used two cardiac fibrosis animal models. Mice developed significant cardiac fibrosis after continuous infusion of Ang II by osmotic pump for 4 weeks and mice with repeated subcutaneous injection of ISO for 2 weeks. Masson’s trichrome staining exhibited a significant cardiac fibrosis area (Figure 1A,B and Appendix A). We also evaluated the blood pressure of mice treated with Ang II and found that Ang II infusion for 4 weeks significantly increased the mean blood pressure (Appendix A). During the search for genes that are differentially regulated in cardiac fibrosis, we collected the ventricle samples and proteins were used for TMT-based quantitative proteomic analysis. We found that Wdr5 was significantly upregulated after Ang II treatment (Figure 1C).

To determine whether the expression of Wdr5 is altered in Ang-II-infused mice, we assessed the Wdr5 mRNA and protein expression in ventricle muscle and identified upregulation of Wdr5 in mRNA and protein by RT-qPCR and western blot. The level of col1a1 and α-SMA was also upregulated both in mRNA and protein in Ang-II-infused mice (Figure 1D–F). We also observed increased H3K4me3 in the Ang-II-infused mice (Figure 1E). Immunohistochemistry (IHC) results also confirmed that Wdr5 positive cell ratios were increased in Ang-II-infused mice (Figure 1G,H). We postulated the increased expression of Wdr5 was mainly due to fibroblasts. To further validate the relationship between Wdr5 and cardiac fibroblasts, another group of mice was subjected to repeated subcutaneous injections of ISO for 2 weeks to induce cardiac fibrosis. Mice with ISO injection developed severe cardiac fibrosis, with increased col1a1 and α-SMA both in mRNA and protein levels (Appendix A). Consistent with our previous results, we found the mRNA and protein of Wdr5 were also increased in mice treated with ISO. We concluded that Wdr5 was upregulated during the cardiac fibrosis pathological process.

### 3.2. Wdr5 Is Upregulated during Ang-II-Induced FMT Process

In the pathological process of cardiac fibrosis and remodeling, FMT-derived myofibroblasts are the major origin of collagen and other ECM proteins [13,29]. We asked whether Wdr5 is responsible for myofibroblast differentiation. We isolated neonatal rat cardiac fibroblasts and stimulated them with different doses of Ang II to induce FMT. Western blot and qRT-PCR analysis indicated that the protein and mRNA levels of α-SMA and col1a1, the molecular markers of FMT, were elevated after Ang II treatment (Figure 2A). The conclusion was also confirmed by the immunofluorescence of α-SMA and col1a1 (Figure 3D and Appendix A). During Ang-II-induced FMT, we observed increased expression of Wdr5 both in protein and slightly in mRNA level, as well as increased H3K4me3. H3K4me3 markers were also increased as shown by immunofluorescence (Figure 2A–E). Our findings indicate that Wdr5 expression and histone H3 lysine 4 trimethylation are upregulated during Ang-II-induced FMT in NRCFs.

### 3.3. Pharmacological Inhibition of Wdr5 Activity Suppresses the H3K4me3 Level and FMT Process

The H3K4me3 methyltransferase is activated by forming a complex with Wdr5 [17,21,24]. There are two overlapping sites on Wdr5 to directly interact with other proteins, which are the “Wdr5-binding motif” (WBM) site and the “Wdr5-interacting” (Win) site. OICR-9429 and MM-401 are known to inhibit the HMT activity of Wdr5 complexes by interacting on its Win site [21,27]. To further validate the role of Wdr5 in FMT, we administered OICR-9429 and MM-401 to neonatal rat cardiac fibroblasts for 48 h before the administration of Ang II. Increased contractile function of fibroblasts was associated with increased expression of α-SMA mRNA and protein, indicating increased myofibroblast differentiation (Figure 3A,B). Cells treated with the Wdr5 inhibitor OICR-9429 and MM-401 did not show downregulation in Wdr5 transcript level, which showed that small molecular inhibitor administration does not interfere with Wdr5 transcript in vitro. However, they inhibited the HMT activity of Wdr5-containing complexes, leading to the decrease of H3K4me3 (Figure 3B). During cardiac fibrosis, differentiated myofibroblasts exhibit contractile fibers containing α-SMA and are responsible for the accumulation of ECM. We further sought to determine whether Wdr5 regulated the process of FMT. Ang II treatment significantly increased the FMT markers, such as α-SMA and col1a1. Western blot and qRT-PCR indicated that α-SMA and col1a1 were downregulated after administration of OICR-9429 and MM-401 (Figure 3A–C). Immunofluorescence analysis of α-SMA and col1a1 in fibroblast also confirmed the inhibition of myofibroblast differentiation by OICR-9429 (Figure 3D,E and Appendix A). Another key property of cardiac fibrosis and FMT is the increased migratory ability of fibroblasts following cardiac injury. The effect of Wdr5 inhibition of cell migration was evaluated by performing transwell and scratch-wound healing assays. OICR-9429-treated cells displayed a significant reduction in Ang-II-induced fibroblast migration (Figure 3F,G). 

### 3.4. Genetic Inhibition of Wdr5 Activity Suppresses the H3K4me3 and FMT Process

The existent inhibitors of Wdr5 generally inhibited the Win site alone, without reducing the expression of Wdr5. However, the Wdr5 inhibitor cannot reduce the expression of Wdr5. To further investigate if Wdr5 is required for the activation of Ang-II-induced FMT, we used the Wdr5 loss-of-function approach involving Wdr5 silencing with shRNA. First, we designed three shRNAs that targeted Wdr5, transfected these shRNAs into fibroblast, and confirmed gene knockdown by qRT-PCR at 48 h post-transfection. We used lentiviruses to transfect the sh-Wdr5 into primary fibroblasts and confirmed that sh-Wdr5(2) was the most efficient in knocking down Wdr5 mRNA by up to 60% (Appendix A). We confirmed the knockdown efficiency and found that it could knock down the Wdr5 mRNA level by up to 60% and the Wdr5 protein level by up to 40% (Figure 4A–C). 

We found that transfection of the lentivirus carrying sh-Wdr5(2) indeed reduced the expression of Wdr5 and its methyltransferase activity. Immunofluorescence results demonstrated that fibroblasts exhibited significantly higher levels of α-SMA than control after Ang II stimulation. Transfection with sh-Wdr5 significantly attenuated the Ang-II-induced changes in α-SMA and col1a1 expression in fibroblasts (Figure 4A–C and Appendix A) but did not reduce the mRNA transcription of ECM components such as FN1 and CTGF (data not shown). Immunofluorescence analysis demonstrated that genetically knockdown Wdr5 depressed the expression of α-SMA and col1a1 (Figure 4A–C and Appendix A). Therefore, we found that silencing Wdr5 achieved the same effect as OICR-9429 and MM-401 in inhibiting the FMT process. Finally, we also evaluated the effect of Wdr5 inhibition in cell migration. Transwell assays and scratch-wound healing assays were also performed, and the results showed genetical inhibition of Wdr5 alleviated the cardiac fibroblast migration as the same effect as OICR-9429 (Figure 4D,E).

### 3.5. Wdr5 Inhibition Attenuated NRCF Proliferation and Induced Cell Senescence

The cardiac fibroblast promoted its proliferation ability under injury or stimulation. To evaluate the effect of Wdr5 inhibitions on NRCFs proliferation, we used Edu incorporation, CCK8 assay, and cytometry assay. We both applied pharmacologic and genetic inhibition of Wdr5. After 72 h treatment of OICR-9429 or 48 h transfection of sh-Wdr5 by lentivirus, we tested the proliferation of NRCFs. OICR-9429 and sh-Wdr5 inhibited the incorporation of Edu in NRCF compared with each control group (Figure 5A,B). Cell cycle analysis using propidium iodide staining and cytometry assay revealed that inhibition of Wdr5 significantly decreased the percentage of cells in G2/M phases (Appendix A). The results of CCK-8 were also consistent with that. By inhibition of Wdr5, the cell viability was decreased compared to control groups in 48 h (Figure 5C,D). According to these, we conclude that Wdr5 inhibition resulted in attenuated NRCFs proliferation. 

It is generally known that cell senescence is related to cell cycle arrest. Many studies reported that Wdr5 is also associated with cell senescence markers such as p21 and p16 in cancer cells [24,30,31]. Therefore, we wonder whether Wdr5 inhibition showed an effect on fibroblasts’ senescence. We applied senescence-associated β-galactosidase (SA-β Gal) staining and used RT-qPCR to evaluate gene expression of senescence-associated secretory phenotype (SASP). The SA-β gal activity was increased after Wdr5 inhibition (Figure 5E). Gene transcription of senescence-associated secretory phenotypes, such as IL-1β, IL-6, MCP-1, TNF-α were significantly higher after inhibition of Wdr5 (Figure 5F). These results confirmed that Wdr5 regulated the proliferation and senescence of cardiac fibroblasts.

### 3.6. Wdr5 Inhibition Affected the Mdm2/P53/P21 Pathway to Induce Cell Senescence

P53 is an important negative regulator of the fibrotic process [10,13,32] and an increased level of p53 is proved to suppress fibroblast proliferation, which was validated in our study. Many studies showed that inhibition of Wdr5 resulted in translational stress and stimulated p53 protein synthesis [26,27]. We wonder whether inhibition of Wdr5 in cardiac fibroblasts affects the p53/p21 pathway and FMT process. Inhibition of Wdr5 showed a remarkably decreased H3K4me3 level. Importantly, genetically and pharmacologically inhibition of Wdr5, significantly increased p53 protein expression, but not in the mRNA level (Figure 6A–F). Therefore, we presumed the regulation of p53 was due to post-translational modification (PTM). It is generally known that Mdm2 induces p53 protein degradation and inhibition of Wdr5 upregulated p53 expression and a previous study and Wdr5 reduced wild-type p53 protein expression by inducing H3K4me3 at the Mdm2 gene promoter [33]. We found Wdr5 inhibition downregulated Mdm2 in NRCFs both in mRNA and protein level (Figure 6A–F). As an important downstream of p53, p21 protein was also upregulated in transcription and translation level (Figure 6A–F), which indicated NRCFs senescence. We also transfected recombinant lentivirus containing Mdm2 to NRCFs to overexpress the Mdm2 level after inhibition of Wdr5. We found that forced overexpressed Mdm2 rescued the upregulated P53 and P21 levels after inhibition of Wdr5 (Figure 6A–F).

In our study, we proved that the Mdm2/p53/p21 pathway was, at least, partially downstream of Wdr5 inhibition in cardiac fibroblast. Inhibition of Wdr5 decreased the level of mRNA and protein levels in Mdm2 and therefore promoted p53 and finally increased the transcription and expression of p21. 

It is generally admitted that Wdr5 is a core subunit of histone H3K4 methyltransferase, leading to transcriptional activation of target genes. A previous study proved that Wdr5 regulated the transcription of Mdm2 in neuroblastoma via affecting the level of H3K4me3 at its promoter [33]. To understand whether Wdr5 is essential for H3K4me3 in Mdm2 gene promoter in cardiac fibroblasts, we next performed Cut&Tag assay with IgG and H3K4me3 antibody, followed by qPCR or PCR with primers targeting the Mdm2 gene promoters (Figure 6G). We proved that H3K4me3 was enriched at Mdm2 promoter by PCR assay using primer A (Figure 6H). We next use OICR-9429 and shWdr5 to inhibit Wdr5 activities and then evaluated the H3K4me3 enrichment at the Mdm2 promoter. Cut&Tag-qPCR analysis showed a significant decrease of the enrichment of H3K4me3 at the Mdm2 promoter in cardiac fibroblasts after Wdr5 inhibition (Figure 6I,J), suggesting that Wdr5 is required for H3K4me3 at Mdm2 gene promoter. Our results showed that Mdm2 promoter activity was downregulated by Wdr5 activity inhibition.

### 3.7. Pharmacological Inhibition of Wdr5 Prevents Ang-ΙΙ-Induced Myocardial Fibrosis

To confirm these findings in vivo, we examined the effect of OICR-9429 in a murine model of cardiac fibrosis induced by Ang II infusion. We administered Ang II via osmotic mini-pump for 4 weeks, followed by an intraperitoneal injection of OICR-9429 (3 mg/kg) beginning the 1st day after Ang II treatment. The dose of OICR-9429 has been previously shown to inhibit the activity of Wdr5 without significant toxicity in mice [34]. We next confirmed that OICR-9429 injection inhibited the expression of H3K4me3. As previously described, Wdr5 inhibition led to fibroblast senescence. To validate these findings in vivo, we treated mice with Ang II and with or without OICR-9429 injection. The hearts were co-stained with P21 and the fibroblast marker vimentin [11]. Immunofluorescence of p21 in the OICR-9429 group exclusively localized in vimentin-positive cells and p21 expression were upregulated after application of OICR-9429 (Figure 7A,B). These data indicated that Wdr5 inhibition increased selective cardiac fibroblast senescence.

We further evaluated the respective effects of OICR-9429 on Ang-II-induced and ISO-induced cardiac fibrosis. Masson’s trichrome staining demonstrated that interstitial, perivascular, and atrial fibrosis were attenuated after Wdr5 inhibition (Figure 7C,D). We also confirmed that profibrotic genes such as col1a1 and α-SMA were also downregulated, which showed impaired fibroblast-to-myofibroblast transition. To testify whether the anti-fibrotic effect was achieved by changing blood pressures, we applied a non-invasive tail-cuff to evaluate the blood pressure of each group. Wdr5 (3 mg/kg) inhibition did not affect the blood pressure of mice (Appendix A). Finally, we found that OICR-9429 did not affect ventricular function by measuring ejection fractions and fractional shortenings (Figure 7E,F). Taken together, we concluded that pharmacologic inhibition of Wdr5 could inhibit the activation of cardiac fibroblast via inducing cell senescence and finally attenuate cardiac fibrosis.

## 4. Discussion

In our study, we discovered that the expression of Wdr5 and the activity of H3K4me3 were upregulated in cardiac tissue of the cardiac fibrosis models of mice involving the subcutaneous infusion of Ang II and injection of ISO. Moreover, for in vitro experiments, administration of the MLL/Wdr5 complex inhibitors, or genetically knockdown of Wdr5 in neonatal cardiac fibroblasts decreased H3K4me3 levels; therefore, inducing cell senescence and inhibiting fibroblast-to-myofibroblast transition. Wdr5 inhibitor, OICR-9429, also selectively induced cell senescence in cardiac fibroblasts with increased P21 and suppressed cardiac fibrosis in Ang-II-infused mice. Finally, we showed that inhibition of Wdr5 decreased the H3K4me3 enrichment at the promoter of Mdm2. The results indicated that upregulated H3K4me3 can promote Mdm2/P53/P21 axis, which was essential in cardiac fibroblast-to-myofibroblast transition (FMT).

Cardiac fibrosis is commonly regarded as the most remarkable structural remodeling in heart failure, myocardial infarction, and atrial fibrillation [6,35]. Excessive deposition of extracellular matrices (ECM) proteins such as collagen and fibronectin are the main contributor to the pathological process of cardiac fibrosis. Studies found that ECM accounts for approximately 25% of the mass in the murine heart [36]. Many types of cells especially fibroblasts are significantly associated with the synthesis and maintenance of ECM. In response to various cardiac stress or damage, including inflammatory cytokines, growth factors, and physical stretch, cardiac fibroblasts can be activated and eventually converse the phenotype to myofibroblasts [37]. Phenotypic features of myofibroblasts included increased production of ECM, expression of α-SMA, and the ability to contract. Angiotensin II induces cardiac fibrosis by the accumulation of fibroblasts and fibroblast-to-myofibroblast transition. It was reported that the inhibition of FMT by different approaches attenuated cardiac fibrosis in Ang-II-infused models [13,38,39]. In Ang-II-infused or ISO-subcutaneously-injected models, we found increased Wdr5 both in mRNA and protein levels. In the Tabula Muris single cell database (https://tabula-muris.ds.czbiohub.org/ (accessed on 15 July 2022)), it showed that endothelial cells and fibroblasts presented higher Wdr5 expression than cardiomyocytes. In our IHC results, the high expression of Wdr5 was found to be concentrated in the interstitial and perivascular areas under high magnification, but in the vessels shown in Figure 1G, we found fewer positive cells in epithelium. The positive cells were mainly found in the interstitium between the heart muscle fibers. Considering the shape, location, and quantity of the cells, we speculated that Wdr5 might be mainly expressed in fibroblasts. We concluded that the Wdr5 activity was enhanced in cardiac fibrosis and then focused on the effect of Wdr5, especially in fibroblasts. Our findings indicated that Wdr5 expression is controlled by the Ang II signaling pathway and is upregulated during Ang-II-induced FMT.

Wdr5 is a highly conserved WD40 repeat-containing protein, and recently emerging evidence demonstrated that it is essential for multiple cellular processes. It formed the COMPASS or called trithorax group (TrxG) complexes with SET1/MLL family of histone methyltransferases. From a structural perspective, Wdr5 contains two major sites that interact with other protein partners: the “Wdr5-binding motif” (WBM) site, and the “Wdr5-interacting” (Win) site. Conclusively, Wdr5 is the core component of these enzymes and it was shown directly associated with methylated histone H3K4, the mark of transcriptionally active chromatin [18,21]. 

Many studies discovered that the regulation of Wdr5 plays an important role in targeting fibrosis genes. It is reported that in TGF-β-treated renal tubular epithelial cells (RTECs), it showed that CTGF proximal promoter abounded with active histone markers (H3K4me3). Congruently, Wdr5 was detected on the CTGF proximal promoter and col1a1 and col1a2 promoters [25,40]. Shimoda et al. found that H3K4me3 was significantly upregulated in the kidney ischemia-reperfusion injury (IRI) model. They also proved that inhibiting the Wdr5 activity attenuated renal fibrosis and renal inflammation in IRI mice [24]. However, no study focused on the Wdr5 in cardiac fibroblasts and its role in cardiac fibrosis. We, therefore, hypothesized that inhibition of Wdr5 could reduce the FMT and alleviate cardiac fibrosis in different animal models and may become a new therapeutic target. By western blot, qPCR, and immunofluorescence, inhibition of Wdr5 resulted in decreased markers of FMT. Similar to Shimoda’s study, these results presented the inhibition of NRCF activation and ECM synthesis following the suppression or knockdown of Wdr5. Taken together, elevated levels of Wdr5 activity and H3K4me3 resulted in fibroblast-to-myofibroblast transition. We also transfected the lentivirus to overexpress Wdr5 in NRCFs. Inconsistent to our hypothesis, it did not promote the FMT. We hypothesized that Wdr5 plays a regulating role through histone3 K4 trimethylation, and the two functional domains (“Wdr5-binding motif” (WBM) site, and the “Wdr5-interacting” (Win) site) bind to different proteins in cell. Therefore, the functional activation of Wdr5 may be a more critical part, while simply increasing the protein expression of Wdr5 may not be sufficient to induce fibroblast activation.

Wdr5 has generally been well-studied in tumor cells. Bryan et al. found that Wdr5 is bound with a conserved set of genes in different types of cancer cells, most of which are those connected to protein synthesis [41]. In many studies, p53 signaling emerged as a common target in Wdr5 knockout or pharmacologically inhibition. Suppression of Wdr5 activity promotes translational stress and stimulates p53 protein synthesis, p53-dependent transcription regulation, and finally controls cell proliferation and fate [27,42]. We wondered whether Wdr5 regulated cardiac fibroblasts via the p53 pathway as well.

The relationship between the p53 protein and cardiac fibroblasts is well illustrated. Upregulation of p53 by Doxorubicin inhibited the proliferation, migration, and cell viability of CFs. Structural genes, such as col1a1 and col4a2 were significantly downregulated [43]. Nagpal et al. discovered that miR-125b is upregulated in the murine model of Ang II infusion and TAC. MiR-125b induced fibroblast proliferation and FMT via inhibition of p53. In other organs, such as lungs and livers [32,44], the inhibition of p53 also attenuated fibroblast proliferation or activation. In the pathology of fibrosis, p53-dependent cell senescence may be an emerging therapeutic target. It was reported that in the model of myocardial infarction (MI), senescent fibroblasts were accompanied by a high level of p53 expression and repressed fibroblast proliferation. Cellular senescence of fibroblasts limited cardiac fibrosis in models of aged, MI, and transverse aortic constriction [10,11,45]. These studies indicated that selectively premature senescence of cardiac fibroblast is an essential antifibrotic mechanism in myocardial fibrosis. In our study, we find similar results. Application of OICR-9429 and Sh-Wdr5 increased the SA-β-Gal positive cell ratio and markers of senescence-associated secretory phenotype (SASP) in NRCF and the mRNA and protein of p21 were also increased in NRCF. To evaluate the proliferation of NRCF, we performed Edu staining, CCK-8 assays, and flow cytometry. Promoted incorporation of Edu, decreased cell viability, and decreased percentage of cells in S plus G2/M phases were observed in NCRF with inhibited Wdr5. These data indicated that suppressed Wdr5 activity resulted in suppressed NRCF proliferation through increased p53, p21 protein, and cell senescence.

Wdr5 regulates gene transcription via inducing histone H3K4 trimethylation at target gene promoters; however, the mechanism through which H3K4me3 is regulated in fibroblast remains unknown. We found Wdr5 regulated p53 protein levels but not mRNA levels; therefore, we hypothesized Wdr5 affected p53 by post-transcriptional modification. As Mdm2 induces p53 protein degradation and inhibition of Wdr5 upregulated p53 expression, we next examined whether Wdr5 regulated p53 protein through modulating Mdm2 expression. By performing Cut&Tag-qPCR assay with 3 different pairs of primers of Mdm2 promoter, we found that inhibition of Wdr5 decreased the enrichment of H3K4 trimethylation at the promoter of Mdm2 compared to the control groups, which is previously indicated in a study of neuroblastoma [33] Therefore, upregulated Mdm2 resulted in increased p53 and p21 expression, and finally led to cell senescence. Taken together, the data showed that genetic and pharmacologic suppression of Wdr5, at least partially, decreased the H4K4me3 at Mdm2 promoter and regulated the Mdm2/p53/p21 axis and finally caused cell cycle arrest.

In the last part of our study, we applied pharmacologic inhibition of Wdr5 and found selective increase of p21 in vimentin-positive cells in vivo as shown by immunofluorescence. It also decreased the cardiac fibrotic area both in Ang-II-infused or ISO-injected models. The application of OICR-9429 did not improve the blood pressure nor the heart function of the Ang-II-infused models, which indicated the inhibition of cardiac fibrosis did not rely on the effect on blood pressure. In our study, we found that Ang II infusion for 28 days did not significantly impair cardiac function. The Wdr5 inhibitor significantly decreased cardiac fibrosis, given the still high blood pressure and the short duration of medication, the effect on cardiac function may not be significant. To evaluate the effect of Wdr5 on cardiac function, we should apply new models, such as transverse aortic constriction (TAC), in the future. Additionally, we have to notice that the SASP mechanism is particularly important in communicating senescence signal from non-cardiomyocytes to cardiomyocytes and could be detrimental in long term.

There exist some shortcomings in our study. ① Considering overexpressing Wdr5 is not equivalent to enhancing Wdr5 activity, we did not apply overexpression of Wdr5 to testify the effect of it in NRCF, which is similar to other studies of Wdr5 in a tumor. ② It is generally known that Wdr5 is a cellular multitasker. It may have more than two dozen primary direct interaction partners [21]. We only verify its canonical function as H3 lysine 4 methyltransferases in regulating Mdm2 expression. We did not estimate the affinity between Wdr5 and numerous elements such as N-Myc, PRC1, or long non-coding RNAs. The role of Wdr5 in these aspects needs further investigation. ③ As Wdr5 is ubiquitously expressed in different types of cells, further investigations into Wdr5 in cardiomyocytes or endothelia are necessary. Endothelial cell senescence leads to impaired vasodilation, causing disorders such as atherosclerosis, HFpEF, or pulmonary hypertension. Intervening of Wdr5 in endothelia only might be harmful and requires further study. ④ For clinical translation in the future, we should have tested the effect of the Wdr5 inhibitor alone on cardiac function and we should evaluate the cardiac diastolic function (HFpEF phenotype) in Ang-II-infused mice. Since OICR-9429 did not affect the cardiac function in Ang-II-infused mice, considering lower expression of Wdr5 in normal mice, we speculated that the Wdr5 inhibitor showed no impairment of function in normal hearts. To evaluate the Wdr5 effect on cardiac function, we believe that further investigation should be tested in heart failure models (such as transverse aortic constriction, TAC). Inhibition of Wdr5 in targeted cardiac fibroblast using AAV approaches, rather than inhibitor compound, might be more appropriate for clinical translation.

Taken together, we identified Wdr5 as a key epigenetic regulator that promotes cardiac fibrosis and FMT. Inhibition of Wdr5 resulted in downregulated FMT, mainly through inhibiting the Mdm2/p53/p21 pathway. Our findings provided new insight into the mechanism of cardiac fibrosis, and targeting Wdr5 in cardiac fibroblasts might be a potential upstream therapeutic option for cardiac fibrosis.

## 5. Conclusions

Wdr5 may be a critical driver in Ang-II-induced cardiac fibrosis. Our studies showed that pharmacological and genetical inhibition of Wdr5 alleviated the Ang-II-induced fibroblast-to-myofibroblast transition in vitro. The administrator of Wdr5 inhibitor attenuated the histological changes of cardiac fibrosis. We also provide evidence that the antifibrotic effect could be associated with cell senescence of cardiac fibroblasts. Therefore, the Wdr5 is a potential target for the treatment of cardiac fibrosis in further clinical trials.

## Figures and Tables

**Figure 1 biomolecules-12-01574-f001:**
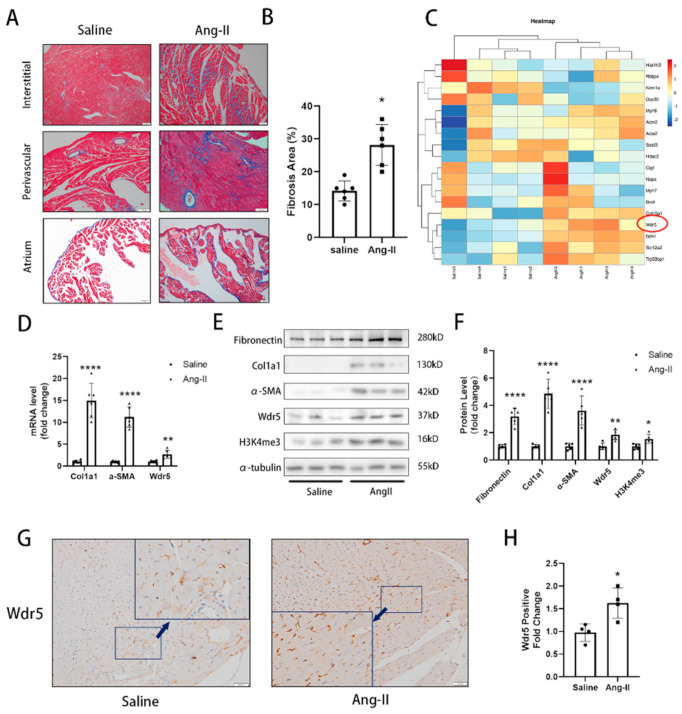
Wdr5 expression was elevated in Ang-II-induced cardiac fibrosis models. (**A**) Masson’s trichrome staining showed increased fibrosis after 28 days of Ang II infusion (Scale bar 100 μm). (**B**) is the quantitative statistic of fibrosis areas in (**A**) (*n* = 5 in each group). (**C**) TMT-based quantitative proteomic analysis of ventricle sample showed increased expression of Wdr5. (**D**) Col1a1, α-SMA, and Wdr5 were significantly elevated in Ang-II-infused animal models in mRNA levels. (**E**) Fibronectin, col1a1, α-SMA, Wdr5, and H3K4me3 protein levels were significantly increased in Ang-II-infused mice compared with saline group (*n* = 5 in each group). (**F**) is the quantitative statistic of (**E**) graph. (**G**) IHC results indicated that Wdr5 were upregulated in the ventricle of Ang-II-infused mice (Scale bar 50 μm). (**H**) is the quantitative analysis of (**G**). Data are presented as mean ± SEM. ***** *p* < 0.05, ****** *p* < 0.01, ******** *p* < 0.0001 vs. saline.

**Figure 2 biomolecules-12-01574-f002:**
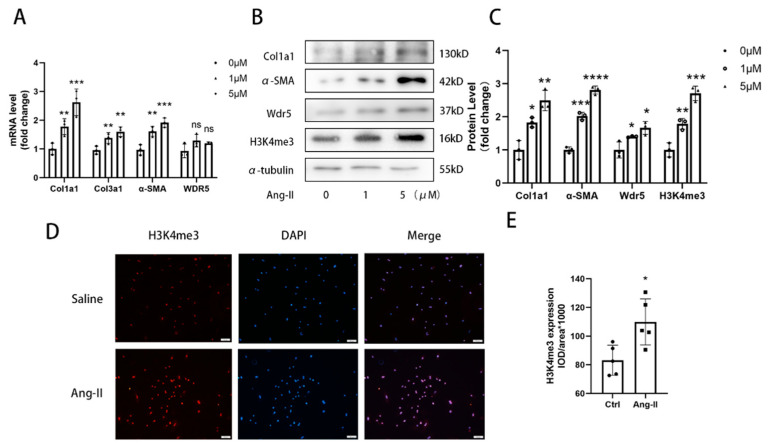
Wdr5 is upregulated in Ang-II-stimulated NRCFs. (**A**) Col1a1, Col3a1, and α-SMA mRNA levels were upregulated after Ang II stimulation. (**B**) Col1a1, α-SMA, Wdr5, and H3K4me3 protein levels were upregulated in respond to Ang II. (**C**) is the quantitative statistic of (**B**) graph. (**D**) Immunofluorescence data of H3K4me3 in NRCFs (Scale bar 100 μm). (**E**) is the quantitative statistic of (**D**) graph. Data are presented as mean ± SEM. ***** *p* < 0.05, ****** *p* < 0.01, ******* *p* < 0.001, ******** *p* < 0.0001 vs. saline.

**Figure 3 biomolecules-12-01574-f003:**
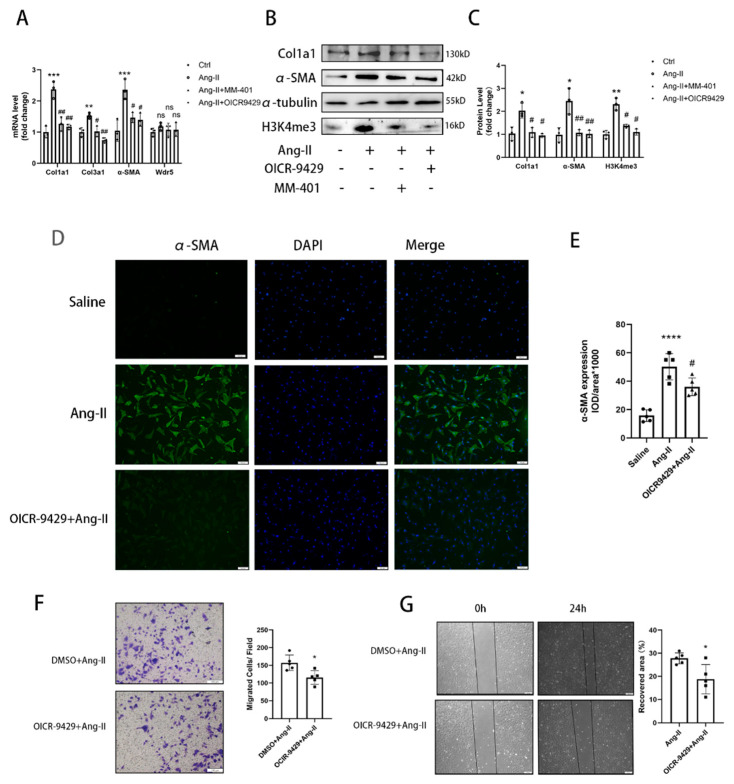
Pharmacologic inhibition of Wdr5 suppressed the FMT process. (**A**) Application of MM-401 and OICR-9429 repressed the Ang-II-induced col1a1, col3a1, and α-SMA. (**B**) Protein levels of col1a1 and α-SMA were downregulated after the application of MM-401 and OICR-9429. (**C**) is the quantitative analysis of graph (**B**). (**D**) Immunofluorescence analysis of α-SMA demonstrated inhibited α-SMA by OICR-9429 (green, α-SMA; blue, DAPI, Scale bar 100 μm). (**E**) is the quantitative statistics of graph (**D**) (*n* = 5 in each group, >3 fields in each sample). (**F**) Representative images of transwell migration assay and the quantification of migrated NRCFs (*n* = 5 in each group, >3 fields in each sample). (**G**) Wound repair assays were performed to assess the migratory capacities of cells that have been treated with DMSO or OICR-9429 (*n* = 5 in each group, >3 fields in each sample). Scale bar 100 μm. Data are presented as mean ± SEM. ***** *p* < 0.05, ****** *p* < 0.01, ******* *p* < 0.001, ******** *p* < 0.0001 vs. saline, # *p* < 0.05, ## *p* < 0.01 vs. Ang II group.

**Figure 4 biomolecules-12-01574-f004:**
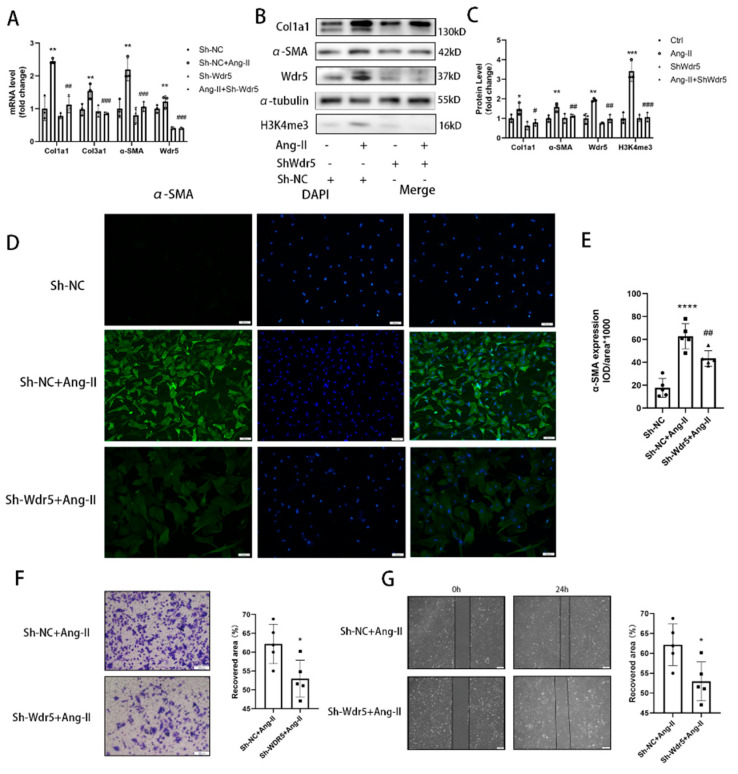
Genetic knockdown Wdr5 suppressed Ang-II-induced FMT process. (**A**) Application of Sh-Wdr5 repressed the Ang-II-induced col1a1, col3a1, and α-SMA in mRNA levels and achieved 60% knockdown of Wdr5 mRNA. (**B**) Protein levels of col1a1 and α-SMA were downregulated after transfection of Sh-Wdr5 by lentivirus (*n* = 4 in each group). (**C**) is the quantitative analysis of graph (**B**). (**D**) Immunofluorescence analysis of α-SMA demonstrated inhibited α-SMA by Sh-Wdr5 (green, α-SMA; blue, DAPI, Scale bar 100 μm) and (**E**) is the quantitative statistics of graph (**D**) (*n* = 5 in each group, >3 fields in each sample). (**F**) Representative images of transwell migration assay and the quantification of migrated NRCFs after transfected with Sh-NC or Sh-Wdr5 (*n* = 5 in each group, >3 fields in each sample). (**G**) Wound repair assays were performed to assess the migratory capacities of cells that have been transfected with Sh-NC or Sh-Wdr5 (*n* = 5 in each group, >3 fields in each sample). Scale bar 100 μm. Data are presented as mean ± SEM. ***** *p* < 0.05, ****** *p* < 0.01, ******* *p* < 0.001, ******** *p* < 0.0001 vs. saline, # *p* < 0.05, ## *p* < 0.01, ### *p* < 0.001 vs. Ang-II group.

**Figure 5 biomolecules-12-01574-f005:**
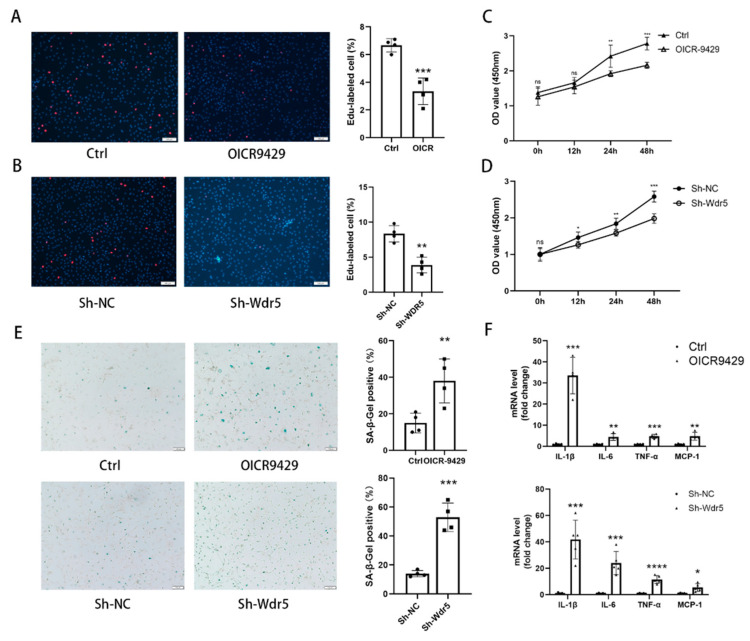
Wdr5 inhibition attenuated fibroblasts proliferation and induced cell senescence. (**A**) OICR-9429 application for 48 h inhibited the Edu incorporation in fibroblasts compared with a control group (*n* = 4 in each group, >3 fields for each sample) (**B**) Sh-Wdr5 infection for 48 h inhibited the Edu incorporation in fibroblasts compared with Sh-NC (*n* = 4 in each group, >3 fields for each sample). (**C**,**D**) CCK8 assay demonstrated that cell viabilities were inhibited after pharmacologic and genetical inhibition of Wdr5 (*n* = 5 in each group). (**E**) SA-β Gal staining of OICR9429 or Sh-Wdr5 treated NRCFs. (**F**) mRNA levels of genes related to senescence-associated secretory phenotype. Scale bar 100 μm. Data are presented as mean ± SEM. ***** *p* < 0.05, ****** *p* < 0.01, ******* *p* < 0.001, ******** *p* < 0.0001 between the 2 indicated groups.

**Figure 6 biomolecules-12-01574-f006:**
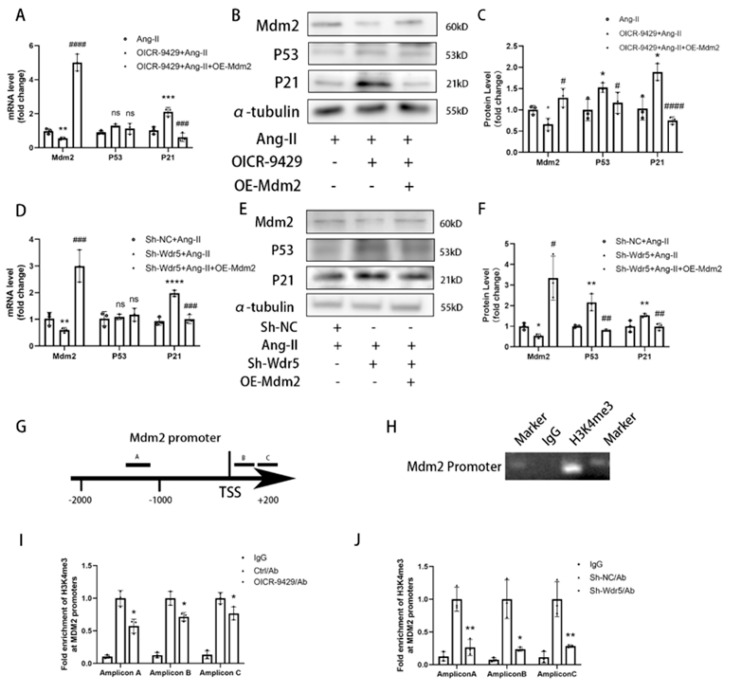
Inhibition of Wdr5 reduced the Mdm2/P53/P21 pathway. (**A**) Application of OICR9429 repressed the Mdm2 and p21 RNA levels. (**B**) Protein levels of P53 and P21 were upregulated after the application of OICR9429 and overexpressing Mdm2 inhibited the increasement. (**C**) is the quantitative analysis of graph (**B**). (**D**) Application of Sh-Wdr5 repressed the Mdm2 and P21 in mRNA levels. (**E**) Protein levels of P53 and P21 were upregulated after transfection of Sh-Wdr5 by lentivirus and overexpressing Mdm2 inhibited the increasement. (**F**) is the quantitative analysis of graph (**E**). (**G**) Schematic representation of the Mdm2 gene promoter containing the H3K4me3 enriched sites. TSS represented transcription start site. (**H**) Cut&Tag-PCR and southern blotting confirmed the enrichment of H3K4me3 on the promoters of Mdm2. (**I**,**J**) Cut&Tag-qPCR indicated the decreased enrichment fo H3K4me3 on the promoters of Mdm2 by 3 amplicons. Data are presented as mean ± SEM. ***** *p* < 0.05, ****** *p* < 0.01, ******* *p* < 0.001, ******** *p* < 0.0001 vs. saline, # *p* < 0.05, ## *p* < 0.01, ### *p* < 0.001, #### *p* < 0.0001 vs. Ang-II group.

**Figure 7 biomolecules-12-01574-f007:**
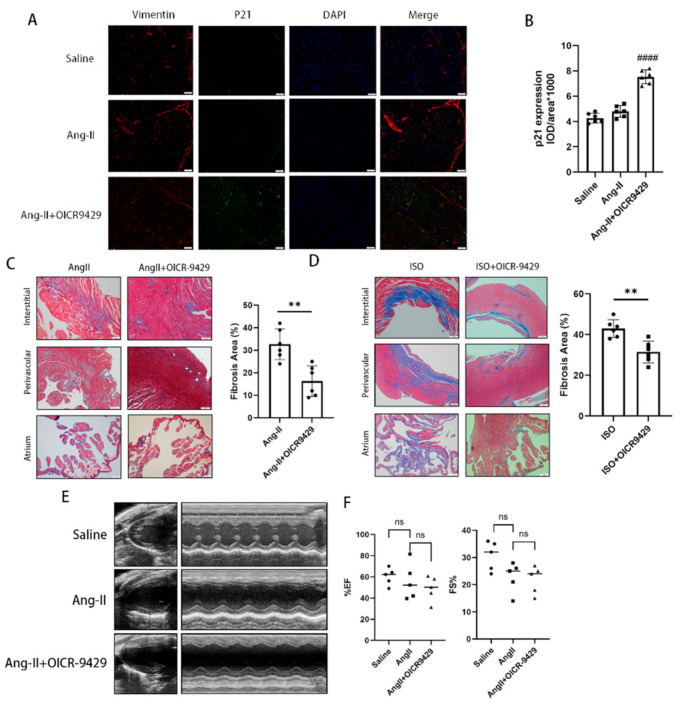
Pharmacologic inhibition of Wdr5 by OICR-9429 induced selective fibroblast senescence and attenuated the Ang-II-induced fibrosis. (**A**) Representative fluorescent images of P21, vimentin (a marker of fibroblasts), and DAPI in heart after treatment of OICR9429 and Ang II (Scale bar 50 μm). (**B**) is the quantitative analysis of graph (**A**). (**C**,**D**) Masson’s trichrome staining showed attenuated fibrosis by treatment of OICR-9429 in the models of Ang II or ISO in different areas. (**E**,**F**) OICR9429 did not affect the ventricular systolic function by measuring EF and FS. Data are presented as mean ± SEM. Scale bar 100 μm. ****** *p* < 0.01, #### *p* < 0.0001 vs. Ang-II group.

## Data Availability

The datasets used and/or analyzed during the current study are available from the corresponding author on reasonable request.

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
