# Peer review of "Inhibition of Wdr5 Attenuates Ang-II-Induced Fibroblast-to-Myofibroblast Transition in Cardiac Fibrosis by Regulating Mdm2/P53/P21 Pathway"

_biomolecules, 2022, doi:10.3390/biom12111574_

Round 1

Reviewer 1 Report

The authors very nicely describe the importance of WDR5 in the process of myocardial fibrosis and its potential implication in fibrosis treatment as well as a promising future target for clinical trials. The authors logically examine the effect of WRD5 mechanism in ECM remodelation, but I have some comments that concern me.

1. In the introduction part some schematic illustration about WRD5 effect is well reccomended

2 . In the material and methods are not pointed the number of experimental rats in groups.

3. How was blood pressure measured?

4. Dilutions of primary antibodies and incubation time in western blot method should be added.

5. It is not very clearly described what part of the heart was used in experiments

6. What kind of microscope was used?

7. Detail information about the statistical evaluation of microscopic pictures is missing. This has to be added.

8. It is very confusing when authors describe in one part results from supplementary files with normal figures. Sometimes is better less.

9. Demonstration of figures and legends is not very well performed. The scale bar of microscopic pictures is missing, pictures are too small

10. Discussion has more character as a conclusion and should be more discuss with other publications.

Author Response

Response to Reviewer 1 Comments

On behalf of my co-authors, we are very grateful to you for giving us an opportunity to revise our manuscript. We have studied reviewers’ comments carefully and revise our manuscript according to the comments.

Thanks again to the hard work of the editor and reviewer!

Point 1: In the introduction part some schematic illustration about WRD5 effect is well recommended.

Response 1: Thank you for your advice. We have added a schematic illustration in supplementary files.

Point 2: In the material and methods are not pointed the number of experimental rats in groups.

Response 2: Thanks to reviewer for reminder, we added the number of mice in each group.

Point 3: How was blood pressure measured?

Response 3: We measured the blood pressure by non-invasive tail cuff. The equipment is Softron blood pressure meter(BP-2010 series). Each mouse required 10 stable BP measurements.

Point 4: Dilutions of primary antibodies and incubation time in western blot method should be added.

Response 4: Thanks to reviewer for reminder, we added the dilutions of primary antibody and incubation time in WB.

Point 5: It is not very clearly described what part of the heart was used in experiments

Response 5: For protein analysis, pathological and IHC analysis, we used ventricle tissues in our experiments. The details have been added to the manuscript.

Point 6: 6. What kind of microscope was used?

Response 6: Olympus BX51 microscope and IX71 (Olympus, Japan) were used in our experiments, and the details have been added to the manuscript.

Point 7: Detail information about the statistical evaluation of microscopic pictures is missing. This has to be added.

Response 7: We are very sorry for our negligence of missing the statistical evaluation of microscopic picture, and the figures have been added.

Point 8:  It is very confusing when authors describe in one part results from supplementary files with normal figures. Sometimes is better less.

Response 8: In order to make the manuscript more clear, we have tried to delete some parts of numbers from supplementary files.

Point 9: Demonstration of figures and legends is not very well performed. The scale bar of microscopic pictures is missing, pictures are too small.

Response 9: We have modified some of the figure legends to make it more clear. We also found the original figures and added the scale bars.

Point 10: Discussion has more character as a conclusion and should be more discuss with other publications.

Response 10: We are sorry that we may have not discussed with other publications, and we have modified parts of the discussion.

Reviewer 2 Report

The manuscript by Yuan et al. reports that the expression of WDR5, a core subunit of H3K4methyltransferase, is increased in cardiac fibrosis and suggests upregulated H3K4me3 promotes cardiac fibroblast-to-myofibroblast transition (FMT) via upregulation of MDM2/P53/P21 pathway. By using two murine cardiac fibrosis models (AngII- and Isoproterenol (ISO)- ) and in vitro neonatal rat cardiac fibroblast (NRCF) culture, the authors showed that WDR5 inhibitors, OICR-9429 and MM-401, as well as WDR5 knockdown, suppressed activation of fibroblasts, FMT, migration, proliferation, and induced fibroblasts’ senescence. They further showed that WDR5 inhibition decreased H3K4me3 at the MDM2 promoter and increased P21 mRNA and protein, which was reversed by MDM2 overexpression. Finally, the authors showed that OICR-9429 attenuated AngII-induced cardiac fibrosis in vivo without altering cardiac functions (EF and FS). The authors claim WDR5 is a potential drug target for cardiac fibrosis; however, reducing fibrosis did not lead to improved cardiac functions. In addition, the long-term effect of cardiac fibrotic senescence on cardiac function is not addressed.

Major points:

1) The long-term effect of WDR5 inhibitors on cardiac fibrosis and cardiac functions should be discussed. Have the authors tested the effect of the WDR5 inhibitor alone on cardiac function?  

2) Methods: line 171 mentioned primary mice atrial fibroblasts, but I could not find how these cells were prepared in the text. Same as primary mice cardiac fibroblasts (line 191). 

3) In Fig. 1G, the legend is missing. It is not clear which cell type expresses H3K4me3 or WDR5. The authors should at least distinguish cardiac fibroblasts and cardiac myocytes. 

4) In Fig. 3D, the magnification seems different. Same for Fig. 4D. 

5) Fig. 3D, did the authors perform control experiments with inhibitor-only conditions?

6) Is WDR5 overexpression sufficient to induce cardiac fibrosis or FMT? Can the authors test this in vitro?

Minor points:

1) Bars are missing in Fig.1A, 1G, 2D, 3D, 3F, 3G, 4D, 4F, 4G, 5A, 5B, 5E, 7A, 7C, 7D and supplemental figures. 

2) Please use either AngII or Ang-II. 

3) Please spell out and abbreviate ISO on line 92.

4) Line 114, info on collagenase type I is missing

5) Please check all the superscripts (for example, line 145, 6*105)

6) Please correct Fig. 6 title (remove the affected)

Author Response

Response to Reviewer 2 Comments

On behalf of my co-authors, we are very grateful to you for giving us an opportunity to revise our manuscript. We have studied the reviewers’ comments carefully and revised our manuscript according to the comments.

Thanks again for the hard work of the editor and reviewers!

Major

Point 1: The long-term effect of WDR5 inhibitors on cardiac fibrosis and cardiac functions should be discussed. Have the authors tested the effect of the WDR5 inhibitor alone on cardiac function?

Response 1: Thank you for your valuable comment. It is a critical issue about the function of WDR5 in a normal heart.

We found that WDR5 expression in normal(wild type) cardiac tissue is not very high, and the expression increased significantly after Ang-II stimulation. In Ang-II stimulated mice for 28days, instead of a decrease in cardiac function, we observed an increase in ejection fraction in some mice. It’s a shortcoming that we didn’t test the effect of the WDR5 inhibitor alone on cardiac function. But since OICR-9429 didn’t affect the cardiac function in Ang-II infused mice, considering a lower expression of WDR5 in normal mice, we speculated that the WDR5 inhibitor showed no impairment of function in normal hearts. We believed that further investigation should be tested in heart failure models (such as transverse aortic constriction, TAC).

Point 2: Methods: line 171 mentioned primary mice atrial fibroblasts, but I could not find how these cells were prepared in the text. Same as primary mice cardiac fibroblasts (line 191).

Response 2: We are very sorry for our negligence of mistakenly writing the methods. Cardiac fibroblasts in our study were neonatal rat cardiac fibroblasts. We have corrected the manuscripts.

Point 3: In Fig. 1G, the legend is missing. It is not clear which cell type expresses H3K4me3 or WDR5. The authors should at least distinguish cardiac fibroblasts and cardiac myocytes. 

Response 3: In IHC results, the high expression of WDR5 was found to be concentrated in the interstitial and perivascular areas of the heart under high magnification, so we speculated that WDR5 might be expressed by non-myocardial cells. In the Genecards database, it showed that WDR5 is mainly located in the Nucleus of cells, which is consistent with our results. In Tabula Muris single cell database, it showed that endothelial cells and fibroblasts presented higher WDR5 expression than cardiomyocytes. Therefore, we hypothesize that non-myocardial cells were the main contributors to WDR5 in hearts. Some comments have been added to the discussion in our manuscript.

Point 4: In Fig. 3D, the magnification seems different. Same for Fig. 4D. 

Response 4: We are very sorry for our negligence. We have found the proper figures and modified the manuscript.

Point 5: Fig. 3D, did the authors perform control experiments with inhibitor-only conditions?

Response 5: We have applied OICR-9429 in a control group of NRCFs, and some results were presented in Western Blot results. For reasons of space of figures, we did not include the results of immunofluorescence. Maybe due to the lower expression in control groups of NRCFs, the inhibitor only didn’t affect the Fibroblast-to-myofibroblast transition.

Point 6: Is WDR5 overexpression sufficient to induce cardiac fibrosis or FMT? Can the authors test this in vitro?

Response 6: Thank you for your valuable comment, it is definitely a critical issue whether overexpression of WDR5 can induce FMT.

When we first designed the study, we decided to knock down and overexpressed WDR5 in fibroblasts and evaluate the effect of them. We also constructed lentiviral vectors coding WDR5 by MV-MCS-EF1α-copGFP-T2A-puro, and we transfected the lentivirus to NRCFs. Inconsistent with our hypothesis, it didn’t promote the FMT. For clinical translation, we then applied pharmacologic inhibitors.

We have reviewed many other studies on the function of WDR5, and few of them intervene by WDR5 overexpression. We hypothesized that WDR5 plays a regulating role through histone3 K4 trimethylation, and the two functional domains (‘WDR5-binding motif’ (WBM) site, and the ‘WDR5-interacting’ (Win) site) bind to different proteins in a cell. So, the functional activation of WDR5 may be a more critical part, while simply increasing the expression of WDR5 may not sufficient to induce fibroblast activation. We have added some conclusions and hypotheses to our discussion parts.

Minor

Point 7: Bars are missing in Fig.1A, 1G, 2D, 3D, 3F, 3G, 4D, 4F, 4G, 5A, 5B, 5E, 7A, 7C, 7D and supplemental figures.

Response 7: We are very sorry for our negligence of missing the scale bars in the figures, we found the original figures and have added the scale bars to the manuscript.

Point 8: Please use either AngII or Ang-II.

Response 6: We have changed the expression to a uniform one.

Point 9: Please spell out and abbreviate ISO on line 92.

Response 9: Thanks for reminding us, and we have corrected the expression as Isoprenaline hydrochloride (ISO, 50mg/kg dissolved in sterile saline) in the manuscript.

Point 10: Line 114, info on collagenase type I is missing

Response 10: Thanks for reminding us, and we have added the information of collagenase type I in the manuscript.

Point 11: Please check all the superscripts (for example, line 145, 6*105)

Response 11: Thanks for reminding us, and we have checked the superscripts carefully in our text.

Point 12: Please correct Fig. 6 title (remove the affected)

Response 12: Thanks for reminding us, and we have removed the wrong word.

Round 2

Reviewer 1 Report

Thank you very much for accepting the recommendations and for a revised improved version of the manuscript. I recommend this article for publishing in MDPI. 

Author Response

It is our pleasure to get your approval. Thank you very much for your efforts

Reviewer 2 Report

This is a revised manuscript by Yuan et al. Although the authors partially addressed this reviewer's previous concerns, some experimental data are unsatisfactory.  

 Fig. 1G. No scale bars are provided in the figure. It is impossible to tell which cell type(s) in the heart express Wdr5. I cannot find perivascular staining of Wdr5 after Ang-II. The authors should perform immunostaining to show that positive cells are cardiac fibroblasts, endothelial cells, or cardiac myocytes. In Fig. 1H, how did the authors quantify and express the data as fold change? Please provide a higher magnification image and present WDR5 positive cells per fibroblasts, endothelial cells, or cardiac myocytes. 

Fig.2. The title says, "WDR5 is upregulated in Ang-II stimulated NRCFs. But the immunostaining only shows H3K4me3. Why don't the authors show WDR5 staining?

Fig. S1. "WDR5 expression were elevated in ISO induced cardiac fibrosis models", but the immunostaining is only for H3K4me3. 

Minor 

1.           Fig. 1A. No information on scale bars. Quantitative statistic of fibrosis areas in A, not D. 

2.           Please use WDR5 or Wdr5 when indicating the protein in NRCF. Please italicize the word to distinguish gene from protein whenever applicable. 

3.           Please check typos and grammatical errors throughout the manuscript. For example, in Fig. 3B, protein levels should be Protein levels. Same for Fig. 4B. In Fig. 4, Genetic knock down instead of Genetical, and Wdr5 should be italicized. Check the title for Fig. 5. 

4.           Fig. 6F, is it the quantitative analysis of graph B?

Author Response

Response to the Reviewer's Comments

On behalf of my co-authors, we are very grateful to you for giving us an opportunity to revise our manuscript. We have studied your comments carefully and revised our manuscript according to the comments.

Thanks again for the hard work of the reviewer!

Major

Point 1: Fig. 1G. No scale bars are provided in the figure. It is impossible to tell which cell type(s) in the heart express Wdr5. I cannot find perivascular staining of Wdr5 after Ang-II. The authors should perform immunostaining to show that positive cells are cardiac fibroblasts, endothelial cells, or cardiac myocytes. In Fig. 1H, how did the authors quantify and express the data as fold change? Please provide a higher magnification image and present WDR5 positive cells per fibroblasts, endothelial cells, or cardiac myocytes.

Response 1: Thank you again for your valuable comments. We applied 1:50 instead of 1:100 antibody to perform IHC again to get better images. We have also changed the images to larger and higher magnification ones.

We can find no WDR5 positive cells in the endothelial in arteries and endocardial cells of the heart (Fig.1G). The positive cells were found in the interstitium between the heart muscle fibers. Considering the shape, location and quantity of the cells, we hypothesized that WDR5 was mainly expressed in fibroblasts.

We quantify the WDR5 expression by image J and a plugin named IHC profiler(https://sourceforge.net/projects/ihcprofiler/). It identified and scored the markers by immunohistochemistry. To diminish this visual perception biasing, IHC profiler has been developed as a standard automated scoring tool. As WDR5 is reported to express major in nucleus, we set a threshold to select all nucleus in the figure, and then used IHC profiler to get the percentage contribution of positive automatically.

Point 2: Fig.2. The title says, "WDR5 is upregulated in Ang-II stimulated NRCFs”. But the immunostaining only shows H3K4me3. Why don't the authors show WDR5 staining?

Response 2: We think that WDR5 regulates transcription by mediating H3K4me3, and the effect of H3K4me3 was more direct. As mentioned, H3K4me2 and me3 are laid down by TrxG, MLL-like, or COMPASS complexes. WDR5 is a core component of these enzymes. The increased expression of H3K4me3 also reflects the upregulation of WDR5 function

We also referred to some other studies related to WDR5, which rarely used immunofluorescence to detect the expression of WDR5. For example:①Kidney Int. 2019;96(5):1162-1175. ②J Hepatocell Carcinoma. 2021;8:333-348. ③Cancer Res. 2015;75(23):5143-5154.

We think this is also a matter of experimental technique. Our antibody (ab178410) is suitable for immunohistochemistry, but not for immunofluorescence. We carried out the immunofluorescence detection before, but could not get the ideal image.

Point 3: Fig. S1. "WDR5 expression were elevated in ISO induced cardiac fibrosis models", but the immunostaining is only for H3K4me3.

Response 3: Thank you for reminding me. We have re-stained the IHC and presented the results of WDR5 expression in ISO models.

Minor

Point 4: Fig. 1A. No information on scale bars. Quantitative statistic of fibrosis areas in A, not D. 

Response 4: Thank you for your notice. We are very sorry for our negligence and we have added the information and changed the figure legends.

Point 5: Please use WDR5 or Wdr5 when indicating the protein in NRCF. Please italicize the word to distinguish gene from protein whenever applicable. 

Response 5: Thank you for your notice. We have changed the expression to formal ones.

Point 6:  Please check typos and grammatical errors throughout the manuscript. For example, in Fig. 3B, protein levels should be Protein levels. Same for Fig. 4B. In Fig. 4, Genetic knock down instead of Genetical, and Wdr5 should be italicized. Check the title for Fig. 5. 

Response 6: Thanks for reminding us, and we have corrected the expression.

Point 7: Fig. 6F, is it the quantitative analysis of graph B?

Response 7: I'm sorry it was a mistake and we have corrected it
